# Update on Classic and Novel Approaches in Metastatic Triple-Negative Breast Cancer Treatment: A Comprehensive Review

**DOI:** 10.3390/biomedicines11061772

**Published:** 2023-06-20

**Authors:** Salvatore Greco, Nicolò Fabbri, Riccardo Spaggiari, Alfredo De Giorgi, Fabio Fabbian, Antonio Giovine

**Affiliations:** 1Department of Translational Medicine, University of Ferrara, Via Luigi Borsari 46, 44121 Ferrara, Italy; salvatore.greco@unife.it (S.G.); riccardo.spaggiari@unife.it (R.S.); 2Department of Internal Medicine, Delta Hospital, Via Valle Oppio 2, 44023 Ferrara, Italy; a.giovine@ausl.fe.it; 3Department of General Surgery, Delta Hospital, Via Valle Oppio 2, 44023 Ferrara, Italy; n.fabbri@ausl.fe.it; 4Department of Internal Medicine, University Hospital of Ferrara, Via Aldo Moro 8, 44124 Ferrara, Italy; degiorgialfredo@libero.it; 5Department of Medical Sciences, University of Ferrara, Via Luigi Borsari 46, 44121 Ferrara, Italy

**Keywords:** breast cancer, triple-negative breast cancer, treatment, systematic review

## Abstract

Triple-negative breast cancer (TNBC) accounts for almost 15% of all diagnosed breast cancers and often presents high rates of relapses and metastases, with generally poor prognosis despite multiple lines of treatment. Immunotherapy has radically changed the approach of clinicians towards TNBC in the last two to three years, even if targeted and specific therapeutic options are still missing; this unmet need is further justified by the extreme molecular and clinical heterogeneity of this subtype of breast cancer and by the weak response to both single-agent and combined therapies. In March 2023, the National Comprehensive Cancer Network (NCCN), the main association of cancer centers in the United States, released the last clinical practice guidelines, with an update on classic and novel approaches in the field of breast cancer. The purpose of this comprehensive review is to summarize the latest findings in the setting of metastatic TNBC treatment, focusing on each category of drugs approved by the Food and Drug Administration (FDA) and included in the NCCN guidelines. We also introduce part of the latest published studies, which have reported new and promising molecules able to specifically target some of the biomarkers involved in TNBC pathogenesis. We searched the PubMed and Scopus databases for free full texts reported in the literature of the last 5 years, using the words “triple-negative breast cancer” or “TNBC” or “basal-like”. The articles were analyzed by the authors independently and double-blindly, and a total of 114 articles were included in the review.

## 1. Introduction

Breast cancer (BC) is the most commonly diagnosed cancer worldwide, and it accounts for 1 in 8 cancer diagnoses; in 2020, more than 2.3 million new cases of BC were diagnosed globally, with about 685,000 deaths directly related to it (source https://www.who.int/news-room/fact-sheets/detail/breast-cancer; accessed on 1 March 2023). About 15% of women who develop BC are diagnosed with triple-negative breast cancer (TNBC) [1], which is the most aggressive form of the disease, with the highest percentages of relapse and/or metastases that frequently make TNBC an unresectable tumor. 

TNBC typically does not express estrogen and progesterone receptors and lacks amplification/overexpression of the human epidermal growth factor 2 (HER2) [2,3], thus hindering targeted and specific therapies. Moreover, TNBC encompasses a heterogeneous group of cancers, and for this reason, its treatment remains among the hardest challenges for clinicians.

For years, and despite high toxicity rates, chemotherapy has remained the standard of care for all-stage TNBC, although no agent has been recognized as the most specific against this subtype of cancer. Conventional chemotherapy, involving mainly anthracyclines and taxanes, is considered the first-line treatment, especially in the pre-operative setting, with a consequent reduction of tumor burden and less extensive demolitive surgery. However, even after chemotherapy treatment, recurrence is common, and prognoses are often poor (source America Cancer Society, updated March 2022, https://www.cancer.org/cancer/types/breast-cancer.html; accessed on 1 March 2023).

Fortunately, in the last 5 years, the treatment paradigm has shifted, allowing for a more tailored approach for patients with TNBC. Due to its high molecular heterogeneity, recent and complex clinical laboratory testing techniques, such as microarray analysis and next-generation sequencing (NGS), have been improved to classify some variants of breast cancer, including TNBC [4]. The hypothesis behind the development of such technologies is that the routine detection of genomic abnormalities (single-nucleotide variants, short insertions or deletions, copy number variations or fusions in multiple genes) could theoretically lead to determining which therapy is the most suitable for a certain TNBC subtype.

Alongside this molecular heterogeneity, TNBC is characterized by an extremely heterogeneous clinical behavior, and, in the last years, several efforts have been made to dissect the complex molecular landscape of this cancer. Histologically, almost all TNBCs (up to 95%) have been classified as invasive ductal carcinomas, while a minority have presented different and varied histological aspects [5]. Several attempts have been made to also establish a molecular classification of TNBC according to transcriptomic and genomic studies. Among the best-known molecular classifications, the one that resulted from the PAM50 (mRNA expression of 50 genes) analysis [6] and the classification by Lehmann et al. [7] must be mentioned. Most TNBCs were classified as basal-like by PAM50 (80.6%) through a direct comparison of 374 cancer samples, followed by HER2-enriched (10.2%), normal-like (4.7%), luminal B (3.5%), and luminal A (1.1%); none of the intrinsic subtypes described differed significantly in terms of the rate of pathological complete response (pCR) or survival after chemotherapy [8]. Lehmann et al., instead, recognized six new and stable TNBC subtypes based on their gene-expression profiles: two basal-like subgroups (BL1 and BL2), two mesenchymal ones (mesenchymal, M, and mesenchymal stem-like, MSL), one immunomodulatory (IM), and one luminal androgen receptor (LAR). In 10% of cases, the subtype of TNBC was defined as unstable (UNS); in this case, a retrospective analysis found significant differences among the subtypes in terms of pCR, with BL1 cancers achieving the best responses to chemotherapy [9]. Appendix A summarizes the characteristics of the different subtypes of TNBC according to PAM50 and Lehmann’s classifications.

Moreover, since breast cancer development causes a huge number of molecular alterations in breast epithelial cells able to modulate immune responses, some studies recently focused on the role of the tumor microenvironment. This is often enriched in infiltrating lymphocytes (TILs), making TNBC the most immunogenic breast cancer subtype. In addition, the predictive and prognostic values of such cells depend on the subtype itself, as shown by Ono et al. in 2012 [10]. The discovery of TILs demonstrated the strong immune-stimulating role of this cancer, making the immune system itself a biomarker for contrasting cancer cells’ growth. In this regard, the genomic and transcriptomic analysis of almost 2000 breast cancers’ architecture demonstrated how the tumors with more extensive lymphocytic infiltration, and thus, with stronger immune and inflammatory responses, were the ones with the most promising outcomes [11].

Different genetic alterations are also commonly found in subjects with TNBC, and these may involve the *BRCA1* and *BRCA2* genes, the upregulation of PI3K-AKT pathway molecules, the trophoblast cell-surface antigen (Trop-2), the presence of androgen receptors (AR) and PD-L1/L2 expression. Each of these theoretically represents a target for newly developed and more precise drugs able to determine significant improvements in TNBC patients’ survival and quality of life.

These genetic alterations underline also defects in the mechanisms of DNA repair, such as homologous recombination (HR), a protective process through which the body is able to substitute a damaged DNA portion using the homologous counterpart as a template. In fact, HR deficiency is common in TNBC and related to *BRCA1* and *BRCA2* loss of function [12]. Therefore, it has been hypothesized that these abnormalities might underpin a role for DNA-damaging compounds as adjunctive therapies [13]. For example, in a subgroup analysis in the study by Zhang et al. [14], the use of cisplatin added to gemcitabine highlighted a better response in HR-deficient individuals.

Because of the unmet need for specific therapies against TNBC and the high rates of metastatic and relapsing diseases immediately related to this subtype of tumor, we aimed to systematically review the literature dedicated to the new therapeutic strategies of the last five years, going beyond the mere classification of the various subtypes of TNBC. We discuss the most recent advances in the field of research, which have determined the compilation of the recent clinical practice guidelines by the National Comprehensive Cancer Network (NCCN), with particular attention given to phase II and III trials.

## 2. Data Analysis

The aim of this study was to analyze, in the updated literature, the existence of new and promising molecular targets in the treatment of metastatic triple-negative breast cancer (mTNBC). We used the population, intervention, comparator, outcome (PICO) model for evaluating the effects of interventions or their variants to indicate outcome comparisons. The search strategy was developed with the assistance of an information specialist.

The topic of this comprehensive review was focused on finding “any evidence useful to suggest and define novel molecular targets to challenge the aggressive character of TNBC”. To date (May 2023), other revisions dealing with the same topic (or a largely similar one) are available. However, a systematic revision was considered useful for the purpose of updating any previously published systematic or non-systematic revisions. mTNBC belongs, in fact, to a scientific area where the scientific progress evolves fast and new clinical trials are already in progress with the need for periodic updates. In this review, all Food and Drug Administration (FDA)-approved antineoplastic drugs included in the last NCCN clinical practice guidelines are discussed.

We searched the PubMed and Scopus databases for free full texts reported in the literature of the last 5 years including the words “triple-negative breast cancer” or “TNBC” or “basal-like”. In the search analysis, we included any of the following studies: multicenter study, observational study, clinical study, clinical trial, (phase I, II, III, or IV), comparative study, consensus development conference, review, systematic review, guideline, meta-analysis, preprints, studies involving patients older than 18 years. The literature search was conducted according to PRISMA guidelines.

A total of 973 articles in PubMed and 200 articles in Scopus were further analyzed by the authors independently and double-blindly, selecting those that contained in the title or in the abstract the following words: “metastatic” or “relapse” or “metastasis” or “metastases”. Each study focused on the theme of therapeutic strategies for mTNBC. In the case of discordance between the first two authors’ opinions, a third opinion was given by another author. After all revisions, a total of 114 articles (98 PubMed and 16 Scopus articles) were included in the review. Figure 1 summarizes the design of the study and the article selection.

### 2.1. Study Selection

Two reviewers (SG and NF) independently screened the articles retrieved from the literature search in three separate stages: (a) titles, (b) titles and abstracts, and (c) full-text selection. Eventual conflicts were resolved by a third reviewer (AG), who decided on acceptance/rejection.

### 2.2. Data Collection Process

The same two reviewers (SG and NF) analyzed all the characteristics and data of the articles selected. These included the study information, patient and tumor characteristics, intervention intervals, and disease outcomes. Any eventual conflicts between the two reviewers were again resolved by discussion and, in case of further discordance, adjudicated by a third reviewer (AG). It was also planned to contact the corresponding authors of the articles for eventual clarification, if needed.

### 2.3. Risk of Bias Assessment

The risk of bias was assessed independently by two reviewers (SG and NF), and eventual conflicts were resolved by discussion. No automation tools or machine learning techniques were used in this research.

### 2.4. Data Source

The analyzed data were extracted from the PubMed database according to the following search queries: (triple-negative breast cancer) OR (TNBC) AND ((y_5[Filter]) AND (ffrft[Filter]) AND (english[Filter]))) OR (basal-like). Filters: Free full text, clinical study, clinical trial, clinical trial, phase I, clinical trial, phase II, clinical trial, phase III, clinical trial, phase IV, comparative study, consensus development conference, consensus development conference, NIH, controlled clinical trial, guideline, meta-analysis, multicenter study, observational study, practice guideline, preprint, randomized controlled trial, research support, N.I.H., extramural, research support, N.I.H., intramural, research support, non-U.S. Gov't, research support, U.S. Gov't, non-P.H.S., research support, U.S. Gov't, P.H.S., research support, U.S. Gov't, review, systematic review, validation study, in the last 5 years, humans, English, female, male, adolescent: 13–18 years, adult: 19+ years, young adult: 19–24 years, adult: 19–44 years, middle-aged + aged: 45+ years, middle-aged: 45–64 years, aged: 65+ years, 80 and over: 80+ years.

For the Scopus Database, we used the following queries: TITLE-ABS-KEY (“triple-negative” OR “triple-negative”) AND (“TNBC”) AND (“basal-like”), selecting the last five years of human studies.

## 3. Antineoplastic Agents

### 3.1. Chemotherapy

#### 3.1.1. Platinum-Based Chemotherapy

TNBC is chemotherapy-sensitive, and for this reason, chemotherapy has remained for years the standard of care (SOC) for this subtype of breast cancer. Platinum-based chemotherapy is the most known chemotherapy regimen and was evaluated in the context of TNBC treatment because of the fact that TNBC commonly harbors somatic *BRCA* gene mutations, which lead to major susceptibility to DNA-damaging agents, such as platinum drugs themselves [15]. In 2010, Silver et al. showed a pCR rate of 22% among the TNBC population treated with neoadjuvant cisplatin [16], while carboplatin was later evaluated in 2015 by Sikov et al., who showed significantly higher pCR rates in early TNBC patients after an association of carboplatin and chemotherapy [17].

Platinum-based chemotherapy was also considered as a treatment option in cases of mTNBC, unfortunately with limited benefits. In 2015, Zhang et al. found longer progression-free survival (PFS) rates in mTNBC patients treated with platinum chemotherapy compared to non-platinum treatment [18], while some other studies showed that mTNBC, differently from TNBC, was more likely to develop resistance to chemotherapeutic agents [19,20]. In the same setting of mTNBC, platinum-based chemotherapies were tested in some phase II and III trials. In 2018, Zhang et al. compared the effects, in terms of survival, in patients treated with a first-line chemotherapy regimen of cisplatin/gemcitabine (GP arm) and that of paclitaxel/gemcitabine (GT arm) [14] as an extension of the CBCSG006 trial previously published [21]. They found a significant interaction between the homologous recombination (HR) status and the treatment for PFS, and the status of HR deficiency significantly correlated with a higher objective response rate (ORR) and longer PFS in the GP arm than in the GT arm. Moreover, no significant interaction between the germline *BRCA1/2* (*gBRCA1/2*) status and the treatment for PFS was found, but patients with *gBRCA1/2* mutations had prolonged PFS in the GP arm compared to the GT arm and, similarly, a numerically higher ORR.

#### 3.1.2. Taxanes

Among the members of the taxane family, paclitaxel has played a fundamental role since its introduction in the context of antineoplastic treatment. It is able to promote the assembly of tubulin into the microtubules of cell cytoskeletons, preventing their dissociation, and thus, blocking cell cycle progression and mitosis. Its use in the setting of mTNBC was tested by Zhang et al. in 2018 [14] in association with gemcitabine and compared with a combined treatment with cisplatin and gemcitabine again; this trial was already discussed in the previous chapter concerning platinum-based chemotherapy. Again, paclitaxel was tested in 2020 by Schmid et al. [22] in association with capivasertib (a kind of tyrosine kinase inhibitor) in the same cancer setting (this trial is discussed in the appropriate section below).

A second member of this family is nab-paclitaxel (paclitaxel bound to albumin). Yardley et al., in their multicenter phase II trial (tnAcity trial), evaluated the risk/benefit profiles of two experimental arms of first-line treatment for mTNBC with nab-paclitaxel [23]. They evaluated the effects of treatment with nab-paclitaxel in association with carboplatin (nab-P/C) or gemcitabine (nab-P/G) compared to those in association with carboplatin and gemcitabine. The nab-P/C treatment resulted in significantly lower PFS rates than nab-P/G (even with similar trends of 12-month PFS) and a numerically longer ORR. Nab-paclitaxel was tested again in 2022 by Wang et al. in their phase III trial, in association with cisplatin, testing the differences in terms of disease outcomes with a second group treated with gemcitabine and cisplatin [24]. The treatment with nab-paclitaxel and cisplatin significantly increased the ORR and prolonged overall survival (OS), showing a good performance of mTNBC outcomes and acceptable tolerability by the patients.

Similarly to paclitaxel, the third member of the taxane family, docetaxel, has the ability to promote microtubule assembly, thus inhibiting microtubule disassembly. It was already tested in 2009 in a phase III trial by Chan et al. [25] in association with gemcitabine or capecitabine (two antimetabolite antineoplastic agents) in the context of mTNBC patients pretreated with anthracyclines, showing no differences between the two arms and suggesting that both antimetabolites were theoretically useful for association therapy with docetaxel in the mTNBC treatment.

No other recent phase II or III trials have considered docetaxel among the main treatments for metastatic TNBC, while a recent phase III study considered it for early TNBC in association with epirubicin (an anthracycline, discussed below) and cyclophosphamide, showing promising results in terms of disease-free survival [26].

#### 3.1.3. Anthracyclines

Anthracyclines are the most common antitumor antibiotics used in the management of TNBC. They can work by different mechanisms, such as impairing the replication of DNA and mitochondrial function, generating oxygen-free radicals, and activating apoptosis, matrix metalloproteinase, and the immune reaction [25].

Therapeutic regimens containing anthracyclines are often better than regimens containing no anthracycline in terms of disease progression; however, they are associated with greater toxicity and usually no improvements in OS. The most common combined therapies involving anthracyclines include cyclophosphamide 5-fluororacil plus epirubicin or doxorubicin (CAF/CEF) and doxorubicin/epirubicin plus cyclophosphamide (AC/EC).

TNBC treatment is often based on a multiagent regimen, which usually leads to improved outcomes; this is usually true for both the preoperative (neoadjuvant) and postoperative (adjuvant) regimens [27]. The IMpassion031 trial [27] is, to date, the main study evaluating the efficacy and safety of doxorubicin in early-stage TNBC. The researchers assessed the results obtained by combined therapy with atezolizumab (a PD-L1 inhibitor, described below) and nab-paclitaxel followed by an AC regimen as neoadjuvant therapy. In this phase III trial, the patients were randomly assigned (in a 1:1 ratio) with the aforementioned regimen or with chemotherapy and placebo. The neoadjuvant treatment with atezolizumab in combination with nab-paclitaxel and anthracycline-based chemotherapy significantly improved the pCR rates, with an acceptable safety profile.

Liposome-encapsulated doxorubicin (liposomal doxorubicin) is the new formulation of doxorubicin, created with the aim of overcoming the cardiotoxic effects of this molecule. The experience concerning the use of liposomal doxorubicin in mTNBC patients is extremely limited, and the few studies published have rarely reported significant results in terms of PFS, pCR, or OS. Two single-center studies have recently reported experience with this drug regimen in association with gemcitabine: the first resulted in significantly longer PFS and OS rates in heavily pretreated mTNBC patients [28], while the second one found the anthracycline chemosensitivity to be an independent predictive and prognostic factor for mTNBC patients receiving PLD [29].

#### 3.1.4. Anti-Metabolites

In the category of anti-metabolites, capecitabine is one of the main antineoplastic agents. It is prevalently used in colorectal and in breast cancers but was also later approved by the FDA for several other oncological disorders. It is a prodrug of fluorouracil, which is enzymatically converted to 5-fluorouracil (5-FU) in tumor tissue through the activation pathway of thymidine phosphorylase, and thus, it is able to block DNA synthesis, inducing cell death [30]. This oral agent was initially approved in anthracycline and taxane-resistant breast cancer and subsequently approved for use in combination with docetaxel as second-line therapy in metastatic BC, or in combination with small-molecule lapatinib in women with human epidermal growth factor receptor type 2 (HER2)-positive metastatic BC, following progression on trastuzumab-based therapy [31,32].

Capecitabine was also tested in the setting of metastatic BC in 2011 in a phase I/II study by Villanueva et al. [30], in association with cabazitaxel or docetaxel (both belonging to the taxane family), showing it to be a plausible alternative in the case of recurrent BC after anthracycline treatment. Always in the setting of metastatic BC (the study included different BC subtypes), the PROCEED phase III trial by Park et al. compared the combination therapy with capecitabine and irinotecan (a topoisomerase1, Top1, pro-drug inhibitor) to capecitabine alone in HER2-negative BC. We report only the results concerning mTNBC (90 subjects), where the combination therapy significantly improved PFS, while the ORRs were only numerically higher in the same sub-cohort of patients and failed to reach statistical significance [33].

The second member of this drug family is gemcitabine, which is a deoxycytidine-analog antimetabolite and a nucleotide analog that inhibits the synthesis of DNA similarly to capecitabine. It was already tested in the early 2000s in association with paclitaxel in the setting of metastatic BC, showing higher response rates and OS when compared to paclitaxel alone [34]. In 2011, Maisano et al. published their phase II trial [35], where gemcitabine was combined with carboplatin in mTNBC patients pretreated with taxanes, showing an ORR of 32% and promising median times of PFS and OS, concluding that gemcitabine was a reasonable option for mTNBC in the case of advanced lines of treatment.

The other phase II and III trials concerning gemcitabine in the setting of mTNBC by Zhang [14], Yardley [23], and Wang [24] were already discussed in previous chapters.

#### 3.1.5. Microtubule Inhibitors

Eribulin mesylate, belonging to a class of anticancer medication of microtubule dynamic inhibitors, was approved by the United States FDA for patients with metastatic BC who have received at least two prior chemotherapy regimens, including an anthracycline and a taxane in either the metastatic or adjuvant setting. The EMBRACE trial, published in 2011 [36], showed significantly improved survival by 2.5 months compared to treatment of physician’s choice (TPC) in women with 2 to 5 prior lines of therapy. In 2018, Mougalian et al. [37], asserted that, in a real-world analysis, eribulin is often used in a more heterogeneous population than that included in randomized-controlled trials (RCTs) in terms of ethnicity and metastatic burden, without significant reduction in terms of OS, safety, or intolerability. They thus suggested that eribulin mesylate could have a significant impact on the clinical benefit when used as a first- or second-line treatment, not only as a third- or fourth-line, even though appropriate RCTs concerning this aspect of treatment are still missing in the literature.

Belonging to the same family, vinorelbine, a vinca alkaloid with the capability of arresting the cell cycle by acting at the microtubular level on tubulin, appears as another therapeutic choice in advanced TNBC. In a subgroup analysis from a study on breast cancer responses to the metronomic administration of vinorelbine by Liu et al. [38] including thirteen TNBC patients, the authors obtained a mean OS of 24.5 months and stable disease in 23.1%. Another study by Valerio et al. investigated instead its use as a first-line agent together with capecitabine in the setting of mTNBC, showing a median PFS and OS of 7.9 and 29.2 months, respectively [39].

However, the administration of vinorelbine was also tested as a second/third-line agent in patients not responding to prior standard chemotherapy, alone or even combined with immunotherapy. In a small cohort of forty-one patients with mTNBC pretreated with taxanes and/or anthracyclines, the administration of vinorelbine along with a platinum regimen also demonstrated promising results in terms of complete (7.3%) and partial responses (26.8%), stability of the disease (34.1%), with an OS of 18.9 months and PFS of 6.7 months [40]. Another recent study (the NAN trial [41]) by Li et al. highlighted the effect of associating vinorelbine with apatinib (belonging to the tyrosine kinase inhibitor family), obtaining improved PFS and OS (3.9 vs. 2.0 months; 11.5 vs. 9.9 months, respectively). This therapeutic possibility was also analyzed in a phase II trial of vinorelbine and oxaliplatin by Zhang et al. [42], showing a median PFS of 4.3 and OS of 12.6 months.

Similarly, ixabepilone, a semi-synthetic analog of epothilone B acting as a microtubule stabilizer, and therefore, arresting the cell cycle, appears to be of interest in advanced BC patients failing first-line therapy (often taxanes/anthracyclines), either used as a single agent or in combination [43]. A pooled analysis obtained from two phase III trials by Rugo et al. [44] highlighted a prolonged median PFS (4.2 vs. 1.7 months, despite no significant differences observed in the OS) and a better response rate (31% vs. 15%) using ixabepilone in addition to capecitabine rather than capecitabine alone in advanced TNBC patients.

#### 3.1.6. Alkylating Agents

Cyclophosphamide is the main drug belonging to alkylating agents and acts prevalently through the inhibition of protein synthesis driven by DNA and RNA crosslinking [45]. It is a type of nitrogen mustard drug able to work as a co-factor with other antimitotic and antineoplastic agents in several malignancies, including Hodgkin and non-Hodgkin lymphoma, multiple myeloma, chronic lymphocytic leukemia (CLL), neuro- and retino-blastoma, small-cell lung cancer, and sarcoma, while its application has been found in the setting of TNBC in association with other standard chemotherapeutic agents, such as epirubicin or doxorubicin, leading to the activation of DNA damage response (DDR) and making these DNA repair mechanisms good targets for antineoplastic therapy [46].

Most trials involving cyclophosphamide prevalently considered this drug as a neoadjuvant combined therapy, with or without adjuvant therapies. The IMpassion031 trial [27] was already discussed in the chapter concerning treatment with anthracyclines, while a recent study by Anders et al. [47], evaluating the use of a priming dose of cyclophosphamide prior to anti-PD1, found that this combined therapy was not effective in ameliorating PFS nor in decreasing peripheral blood regulatory T cells (and thus increasing the antitumoral response, as shown in some pre-clinical observations [48]).

#### 3.1.7. Antibody-Drug Conjugates

Recently, sacituzumab govitecan (SG) also joined the family of chemotherapies tested for mTNBC. This is an antibody–drug conjugate composed of an antitrophoblast cell-surface antigen 2 (Trop-2) IgG1 kappa antibody coupled to SN-38, the active metabolite of irinotecan and a topoisomerase inhibitor [49]. The ASCENT trial [50], published in 2021, demonstrated that PFS and OS were significantly prolonged in the cohort of patients treated with SG compared to single-agent chemotherapy of the physician’s choice (eribulin, vinorelbine, gemcitabine, or capecitabine). All patients had previously received taxanes as a first-line treatment, and all of them were initially diagnosed with mTNBC. In 2022, O’Shaughnessy and colleagues [51], assessed a sub-analysis of the same ASCENT trial, showing how SG could be considered a favorable and manageable therapeutic option for mTNBC regardless of the cancer subtype at the initial diagnosis, and thus allowing for, at least theoretically, the optimal treatment allocation at the first diagnosis.

Based on the concept that almost 60% of human epidermal growth factor receptor 2 (HER2) negative metastatic BC express low levels of HER2, one possible targeted therapy against this subtype of cancer involves HER2-low tumors. This is defined in cases with a score of 1^+^ in immunohistochemical (IHC) analysis or an IHC score of 2^+^ and simultaneous negative results following in situ hybridization (ISH) [52,53]. Since for patients with hormone receptor-negative and HER2-negative metastatic BC, especially in cases of absent *BRCA1/2* mutations, only a few targeted agents are available, a new antibody–drug conjugate was recently developed, trastuzumab deruxtecan, specifically directed towards HER2. This is a humanized monoclonal antibody linked to a topoisomerase I inhibitor, approved by the FDA for the treatment of metastatic HER2-positive BC [54]. It has already been tested in phase I and II studies, with good results in terms of the ORR and PFS in pre-treated patients with HER2-low mBC [55,56], while a very recent phase III clinical trial (DESTINY-Breast04) [57] compared the effects of this anti-HER2 compound in patients randomly assigned (in a 2:1 ratio) to trastuzumab deruxtecan or the physician’s choice of chemotherapy, finding significantly longer PFS and OS in the cohort of subjects treated with the new compound in HER2-low mBC.

### 3.2. Immune Checkpoint Inhibitors

Immune checkpoints refer to a multitude of inhibitory mechanisms involved in the extremely complex immune response to cancer. Such checkpoints are composed of the ligands on the cancer cells and the complementary receptors on the CD8^+^ T cell, and the most known bindings include PD-1/PD-L1, CD80-CD86/CTLA4, MHC II/LAG3, CD155/TIGIT, GAL9/TIM3, and others [58].

Targeted inhibition of such inhibitory molecules has dramatically modified the strategies for activating anti-tumor immunity in cancer therapy, and some humanized antibodies targeting these immune checkpoints have also been tested in the setting of mTNBC. The extremely complicated interactions between T-cells and cancer cells are synthesized in Figure 2.

Gene expression and clinical data analyses of signaling processes involved in TNBC have in fact shown that higher immune response levels are associated with better clinical outcomes [7], while treatment with anthracyclines is able to induce the immune response through the activation of CD8^+^ T cells [59].

Signaling through the programmed death 1 (PD-1) inhibitory receptor upon binding PD-L1 on the cancer cell surface acts like the other molecules cited above, and it is for sure the most studied binding concerning immune checkpoint inhibition. The existence of PD-1/PD-L1 was discovered in the early 2000s by Dana-Farber Cancer Institute scientists in Boston, and nivolumab was the first developed PD-1 inhibitor. It is currently used in certain types of cancers, such as metastatic melanoma, non-small-cell lung cancer (NSCLC), Hodgkin lymphoma, and others, but it is not recommended by the NCCN clinical practice guidelines.

Atezolizumab is an engineered humanized IgG1 monoclonal antibody able to target the PD-L1 protein, thus preventing binding with PD-1 (but allowing, at the same time, the alternative ligand PD-L2 to bind to PD-1 in order to reduce autoimmune hyper-responsiveness) and was tested in several recent clinical trials in the setting of mTNBC. In 2018, Schmid et al. in their Impassion130 phase III trial [60], compared combined treatment with atezolizumab and nab-paclitaxel to placebo and nab-paclitaxel, finding longer PFS and higher ORRs (56% vs. 49%) in the group treated with atezolizumab. Adams et colleagues [61] evaluated the safety, efficacy, and clinical activity of combination therapy with atezolizumab and nab-paclitaxel after a two-year follow-up. The ORR was 39.4%, and the median duration of response (DOR) was 9.1 months; the median PFS and OS were 5.5 and 14.7 months, respectively. Atezolizumab was also tested in another phase II study (the ALICE trial), in combination with immunogenic chemotherapy (pegylated liposomal doxorubicin plus cyclophosphamide) in patients with mTNBC. The association therapy was found to be safe and well-tolerated and led to improved PFS (4.3 vs. 3.5 months compared to the group treated with immunogenic chemotherapy plus placebo) [62]. The latest phase III trial, performed by Miles et al. and published in 2021 (the IMpassion131 trial [63]), evaluated atezolizumab in the same setting of mTNBC, but in association with paclitaxel, comparing the effects of this combined therapy with paclitaxel alone. The patients were randomly assigned to combined therapy or placebo plus paclitaxel in a 2:1 ratio. The PFS and OS analyses did not show significant differences between the groups, while numerically better ORRs were reported in the PD-L1-positive population only.

Another immune checkpoint inhibitor (ICI) is pembrolizumab, which is a humanized IgG k antibody targeting PD-L1, originally developed for treating metastatic melanoma, and tested in the last several years in different clinical trials, also in the setting of mTNBC. In 2019, Adams et al. [64] evaluated the safety and clinical response in patients with mTNBC treated with first-line monotherapy with pembrolizumab. They found a median PFS and OS of 2.1 and 18.0 months, respectively. The ORR, instead was 21.4%. In conclusion, pembrolizumab had a manageable safety profile and durable antitumor activity as a first-line therapy against mTNBC. Ho et al., in 2020, tested pembrolizumab in association therapy with radiotherapy in a small cohort of patients with mTNBC not tested for PD-L1 expression [65]. The ORR for the entire cohort was 17.6%, and the six-month PFS was 18%, even though the main limitation of the study was the small size of the sample considered.

Another important trial concerning pembrolizumab was published in the same year by Schmid et al. [66], with their phase III trial KEYNOTE-522, even though this study was conducted on patients with stage II or III TNBC and not with a diagnosis of mTNBC. Meanwhile, the KEYNOTE-119 trial, published in 2021 [67] was focused on mTNBC patients specifically. Pembrolizumab or single-drug chemotherapy of the physician’s choice (capecitabine, eribulin, gemcitabine, or vinorelbine) were randomly assigned (in a 1:1 ratio) to patients who had received one or two previous systemic treatments. However, despite the promising results derived from previous studies concerning pembrolizumab in the setting of mTNBC, this trial did not show significant improvements in terms of OS, although the authors concluded that pembrolizumab should be reserved for PD-L1-enriched tumors and, preferably, in combination therapies. Based on these observations, in the KEYNOTE-355 trial, Cortes and colleagues [68] evaluated the effects of pembrolizumab plus the investigator’s choice of chemotherapy (nanoparticle albumin-bound paclitaxel, paclitaxel, or gemcitabine–carboplatin) in comparison with placebo plus chemotherapy, finding that the mTNBC subjects with a higher expression of PD-L1 (combined positive score, CPS ≥ 10) obtained better results with combined therapy of pembrolizumab plus chemotherapy than chemotherapy alone.

Dostarlimab-gxly is one of the newly developed ICIs. It is a monoclonal IgG4 antibody targeting PD-1 and was recently approved by the FDA for all subtypes of breast cancer presenting mismatch repair deficiency. The latest NCCN guidelines have introduced this drug among those suggested for third- or higher-line treatment, considering their favorable effect on treating mismatch repair-deficient (dMMR) advanced-stage breast cancer in case of the unavailability of other treatment options [69]. In this sense, although no RCTs concerning dostarlimab-gxly exist, patients with mTNBC who are candidates for treatments including one ICI should undergo testing for mismatch repair/microsatellite instability [70].

### 3.3. PARP Inhibitors

Behind developing carcinogenesis, DNA damage probably represents the leading process and can occur through several mechanisms. Single-strand breaks (SSBs) and double-strand breaks (DSBs) are represented by damage at one or two of the DNA strands, respectively. While SSBs are common and efficient processes, DSBs are comparatively rare. Other DNA damage mechanisms involve helix distortion and replication errors. SSBs, helix-distorting damage, and replication errors are corrected by base excision repair, nucleotide excision repair, and mismatch repair, respectively. On the contrary, DSBs are considered among the most cytotoxic mechanisms of DNA damage, and a key role is played by homologous recombination and non-homologous end-joining (NHEJ) [71,72].

Poly(ADP-ribose) polymerases (PARPs) include several multifunctional enzymes involved not only in base excision repair mechanisms but also in DSB repair [73]. Among them, PARP-1 is the most important one, being essential for maintaining genome integrity [74]. Approximately 25% of patients with TNBC are carriers of breast cancer susceptibility gene 1 or 2 (*BRCA1/2*) deleterious mutations, which are, in turn, essential components of homologous recombination repair (HRR) [75]. For this reason, tumors with *BRCA1/2*-inactivating mutations are strongly dependent on the SSB repair pathways, thus resulting in an accumulation of DNA alterations (and apoptosis) in cases of DNA damage repair impairments [76]. In Figure 3, the mechanism of action of PARP inhibitors is shown.

PARP inhibitors (PARPi) were developed with the aim of avoiding this tumoral cell repair system, leading to the accumulation of unpaired damages and, thus, to tumor cell death. This is obviously true in cases of tumors harboring HRR pathway defects, which are tumors sheltering *BRCA1/2* mutations. Two PARPis were recently approved by the FDA for mTNBC treatment (olaparib and talazoparib), while other members of the same family are still under investigation with the same goal.

Olaparib, an oral member of the PARPi family, was first approved for the treatment of patients with recurrent ovarian cancer and evidence of a *BRCA* mutation. It was later tested in the context of metastatic breast cancer in patients with germline *BRCA1/2* mutations in two separate trials in 2010 and 2015, showing safety and promising antitumoral activity [77,78]. In 2017, Robson et al. published their phase III trial, in which patients were randomly assigned (in a 2:1 ratio) to receive single therapy with olaparib or standard therapy (with capecitabine or eribulin mesylate or vinorelbine). They calculated a median PFS of 2.8 months longer and a risk of disease progression or death 42% lower in the olaparib-treated group, showing a significant benefit of such therapy in HER-2-negative metastatic breast cancer patients carrying *BRCA1/2* mutations [79]. The second and final part of the study (OlympiAD trial) was published in 2019, and, although it did not demonstrate statistically significant improvement in terms of OS by olaparib treatment compared to standard chemotherapy, it showed a possible meaningful OS benefit among the patients not previously treated with chemotherapy [80].

As for talazoparib, instead, a first phase I trial was published in 2017 by De Bono et colleagues [81]. Talazoparib monotherapy resulted in a 50% RR with an overall 86% clinical benefit rate at 24 weeks in a small cohort of 18 patients with advanced BC and germline *BRCA1/2* mutations. Moreover, a phase II trial (ABRAZO study) by Turner et al. published in 2018 showed that the response rate of patients with previous platinum chemotherapy was 21%, while that registered in the group treated with three or more cytotoxic regimens for advanced BC (not treated with platinum-base therapies) was 37% [82]. The leading study concerning talazoparib treatment was published in the same year by Litton et al., the EMBACA trial [83]. This was a randomized, open-label phase III trial, where patients with advanced BC and a germline *BRCA1/2* mutation were assigned (in a 2:1 ratio) to talazoparib treatment or standard single-agent therapy. These researchers showed that talazoparib treatment was able to induce a longer median PFS (8.6 vs. 5.6 months) compared to the one registered in the control group of patients treated with single-agent therapy of the physician’s choice (capecitabine, gemcitabine, or eribulin). Similarly, the ORR was higher in the talazoparib group (62.6% vs. 27.2%), while in general, significant delays in the time to clinical deterioration were favorable with PARPi treatment.

### 3.4. Tyrosine Kinase Inhibitors

Tyrosine kinase inhibitors (TKIs) consist of agents targeting peculiar enzymatic processes involved in the phases of the cell cycle, and therefore affecting cell proliferation and growth or angiogenesis [84]. This ability allows these molecules to act more specifically compared to conventional chemotherapy, thereby determining theoretically minor side effects and giving them the possibility of being administered in combination therapy [85]. We now discuss several kinases and molecular cascades that have been considered possible targets for the treatment of TNBC.

Trilaciclib, an intravenous reversible inhibitor of CDK4/6, is able to arrest proliferation in healthy cells, such as hematopoietic stem cells and lymphocytes, thereby protecting them from the myelosuppressive effect of the subsequent systemic chemotherapy agent [86] but appearing less effective in cases of TNBC [87]. However, the addition of trilaciclib to combined chemotherapy (with gemcitabine and carboplatin) demonstrated a better overall survival (17.8 vs. 12.6 months) in a phase II study by Tan et al., an effect that seemed to not be influenced by the level of PD-L1 expressed [87]. Based on these promising results, a phase III trial is ongoing by Goel et al. (PRESERVE 2), with the aim of establishing the efficacy and safety of trilaciclib administration in a first-line gemcitabine and carboplatin regimen, stratifying patients according to PD-L1 status [88].

Capivasertib, buparlisib, and taselisib share the capability of acting along the phosphatidylinositol 3-kinase (PI3K)/AKT/mammalian target of the rapamycin (mTOR) signaling pathway, which is often over-expressed in breast cancer [89], determining an increase in cellular motility and proliferation and a reduced effect of cytotoxic agents [90]. The oral agent capivasertib has a role as a PI3K/AKT inhibitor and displayed antitumor activity on a pre-clinical level, especially when associated with taxanes [91]. In a phase II trial by Schmid et al., the association of capivasertib with paclitaxel compared to paclitaxel alone demonstrated in over 140 TNBC patients an increase in the survival rate (PFS 5.9 vs. 4.2 months; OS 19.1 vs. 12.6 months), which appeared to be even more significant when considering patients with specific *PIK3CA/AKT1/PTEN*-mutated tumors (PFS 9.3 vs. 3.7 months) [22]. Buparlisib (BKM120) acts as a PI3K inhibitor as well, and in a phase II study by Garrido-Castro et al., it was administered over four weeks on a cohort of fifty women with metastatic TNBC, obtaining stability of the disease over four months in 12% of the patients (surprisingly, none of them had *PIK3CA-*, *AKT1-*, or *PTEN*-identified mutations), with a median PFS and OS of 1.8 and 11.2 months, respectively [92]. As the specificity of PI3K inhibitors favors their use in combination therapy, a phase IB/II study by Lehmann et al. [93]. investigated the association of taselisib, another PI3K inhibitor, with hormonal therapy in a small cohort of TNBC patients. When administered in combination with enzalutamide, taselisib obtained better responses than hormonal therapy alone (CBR of 35.7% vs. 0%, and median PFS of 3.4 months), with patients expressing androgen receptors showing a better response (CBR 75%).

ENMD-2076 is an orally active kinase inhibitor that acts upon Aurora A and other kinases contributing to the process of angiogenesis, such as VEGFRs and FGFRs, with the effect of inhibiting the neoplastic mitotic activity and angiogenesis [94]. In TNBC, this effect was deepened in a phase II trial by Diamond et al. [95] involving forty-one patients with either metastatic or local disease. The authors found a 6-month CBR of 16.7%, with stability of the disease at 24 weeks for most patients, and partial response for only two of them. As could be possibly expected, they also observed on tumor biopsies a decrease in cellular proliferation and microvessel density, and increased p53 (a regulatory protein that is often mutated in human cancers) and p73 (homolog of p53).

Another member of this family is cobimetinib, a molecule that antagonizes mitogen-activated protein kinase (MAPK)/extracellular signal-regulated kinase (MEK), a cascade that is often dysregulated in BC, leading to immunomodulation and chemoresistance [96], and whose inhibition has been shown to potentiate in vitro the apoptotic effect of taxanes [97]. To further investigate its effect as a combination therapy in vivo, a phase II study by Brufsky et al. was performed, and it showed a potential benefit with a better PFS (5.5 vs. 3.8 months) when using the combination of cobimetinib and paclitaxel rather than paclitaxel alone with placebo, despite not reaching statistical significance in a cohort of about one hundred patients [98].

Further therapeutic targets appear to be of interest when considering therapy with TKI. Larotrectinib and entrecitinib are TRK inhibitors that might have a future role in TNBC therapy. Both drugs appear to have good tumor activity in neoplasms that are TRK fusion-positive, including breast cancer [99], and despite the lack, to our knowledge, of phase II/III trials on the role of these agents on TNBC patients specifically, there are some case reports in the literature of TNBC patients with recognized NTRK fusion alterations displaying optimistic results when treated with these agents [100]. Another molecule to be considered is selpercatinib, an inhibitor of RET that is currently used in the treatment of some solid neoplasms, such as thyroid and lung cancers, but is still underexplored in the setting of TNBC [99,101].

Overall, although some TKIs have more solid evidence than others, it is essential to characterize the molecular aberrations present in TNBC patients in order to be able to offer the right therapy, especially when considering an approach with a selected TKI.

### 3.5. Alternative Therapies

Alternative therapies were recently hypothesized to be useful in patients with TNBC, especially in those with advanced cancer progression and previously treated with SOC therapy. Although TNBC typically does not express estrogen or progesterone receptors (ER and PgR, respectively), one TNBC subtype (the luminal androgen receptor, LAR subtype) appears to be hormonally regulated, and its growth was thought to be driven by signaling led by the androgen receptor (AR) [7,102,103]. However, recent studies have shown how up to half of all TNBCs express Ars (not only the LAR subtype), suggesting that Ars could act as potential targets for BC therapy. Enzalutamide, as a kind of AR inhibitor, has the potential to reduce baseline cell proliferation, anchorage-independent growth, migration, and invasion, and it is able to increase apoptosis in AR^+^-TNBCs [104]. Enzalutamide was tested in the setting of mTNBC by Lehmann and colleagues in 2019 [93]. Its AR inhibition efficacy was tested alone or in combination with taselisib. In the first phase of the study (phase Ib), the researchers determined the maximum tolerated dose of both drugs, while in the second phase (phase II), they randomly assigned patients to a single treatment or a dual treatment, finding that the patients with LAR mTNBC had a better clinical response to the associated therapy.

In the context of AR inhibition, recent studies have focused on the role of seviteronel in AR^+^ tumor cells. Differently from enzatulamide, seviteronel is able to inhibit both cytochrome P450 17α-hydroxylase/17,20 lyase (CYP17 lyase, an enzyme necessary for androgen production [105]) and ARs. Besides the hypothetic role that this double inhibition could have on TNBC cells, some researchers have recently reported that AR is able to act as a mediator of radioresistance in TNBC, making it an interesting biomarker for predicting the eventual response to radiotherapy [106]. In this sense, a group of researchers from the USA recently explored the role of seviteronel in the radiosensitivity of AR^+^ TNBC, finding a favorable effect of this drug in combination with radiotherapy, much different from that reported for combined enzatulamide–radiotherapy treatment [107].

The field of research dedicated to antitumoral activities is in continuous progress, and the interest in newly discovered factors promoting oncogenesis or inhibiting cell apoptosis is always higher [108]. Among the second-messenger molecules, hydrogen sulfide (H_2_S) has been found to participate in many physiological and pathological processes [109,110,111]. H_2_S liberation derives from endogenous production with the need for L-cysteine as a substrate. Endogenous H_2_S and low levels of exogenous H_2_S are able to induce angiogenesis and simultaneously inhibit cell apoptosis, accelerating their vital cycle [112]. Contrarily, H_2_S donors can selectively inhibit cancer cell progression, inducing intracellular acidification and inhibiting the proliferation and metastasis of tumor cells through several intracellular pathways [113,114], although the effect of these substances on animal health is still uncertain. Hence, H_2_S donors are, theoretically, a possible therapeutic solution for all types of tumors, and they have been studied also in the context of TNBC. Recently, Li et al. reviewed the role of these compounds in this subtype of BC, leaving the door open for the possibility of their use in the next future [115].

Some specific compounds have been developed in the last several years to restore p53 protein functionalities. The p53 protein is known for playing a significant role in conserving DNA stability, thus preventing cancer development. In particular, after noticing any DNA damage, this protein is induced and activated, causing cell-cycle arrest [116]. Nevertheless, in the case of excessively extended damage, it leads to cell apoptosis and death. Several cancers (and particularly TNBC) are associated with mutations in the *p53* gene (*mtp53*), located on chromosome 17, with obvious uncontrolled DNA damage repair and missed cell apoptosis; by reverting this mutational process, it is theoretically possible to induce altered cell apoptosis, defeating that subtype of cancer. Based on this idea, several natural and naturally derived compounds have already been studied (or are currently being tested) for their ability to target *mtp53*, especially in animal models, but also for the few side effects caused, differently from classic antitumoral drugs.

*p53* reactivation and induction of mass apoptosis *(PRIMA-1*) was recently identified and tested for its ability to induce conformational *mtp53* changes that facilitate binding to DNA, with the aim of leading to cell apoptosis [117]. *PRIMA-1* is also able to recognize cells expressing *mtp53* and unmutated *p53*, and its effectiveness depends especially on the intracellular *mtp53* protein levels. Moreover, the effects of *PRIMA-1* seem to be potentiated by the addition of a methyl group, yielding *PRIMA-1Met*, called *APR-246*, which is more soluble and more active than *PRIMA-1*. *APR-246* activity was tested in animal models and showed capabilities in increasing the levels of reactive oxygen species (ROS), leading cells to autophagy [118].

Among *mtp53* reactivators, together with other plant-derived nutraceuticals, such as luteolin, curcumin, and members of the flavonoids family, which are currently under investigation in the field of oncology, COTI-2, a third-generation thiosemicarbazone, was recently shown to induce antitumor activity also in the context of TNBC, at least in vitro and in vivo in animal models [119]. Besides its activity on *mtp53* reaction, differently from *PRIMA-1* and *APR-246*, COTI-2 seems to be also effective at targeting rapamycin (fundamental in the mTOR pathways, which regulate cell growth, proliferation, and survival), inhibiting it and activating AMP-activated protein kinase (AMPK) [120].

The use of non-coding RNAs (ncRNAs) is also a theoretically useful therapeutic approach in the treatment of some types of cancers, and this happens because such ncRNAs have the ability to regulate gene expression at both the transcriptional and post-transcriptional levels [121]. Short interfering RNA (siRNA) and microRNA (miRNA) are two of the main ncRNAs, and both have the potential for treating aggressive tumors, such as TNBC. miRNAs are able to regulate gene expression by translational repression or mRNA degradation, while siRNAs can specifically silence genes implicated in cancer pathogenesis. Patisiran and givosiran were the first developed siRNA-based therapies, and miravirsen was the first drug belonging to the miRNA family. Currently, neither miRNA-based nor siRNA-based therapies are approved for the treatment of BC, but a recent review shed light on their potential to act as prognostic biomarkers for different cancers, including TNBC [122].

Cancer cells are notably characterized by an aberrant gene expression, and the acetylation of histone proteins contributes to it [123]. Histone deacetylases (HDACs) are the enzymes able to catalyze this reaction, and their inhibition was thought to be useful in combination therapy for some types of cancer, including TNBC. In preclinical studies, chidamide, an HDAC inhibitor, was found to inhibit both the proliferation and migration of TNBC cells [124,125], and, based on this, a group of Chinese researchers recently tested, in a phase II study, the effectiveness of chidamide in combination with cisplatin in the setting of mTNBC [126]. However, this combined therapy did not lead to significant improvement in the ORR or PFS, although the small size of the sample (16 patients enrolled) should be considered.

In 2003, some researchers from the USA first identified from BC the stem cells that gave start to carcinogenesis and were able to distinguish tumorigenic cells from non-tumorigenic cells [127]. Aldehyde dehydrogenase (ALDH) and CD24/CD44 were recognized as two of the main cancer stem cell (CSC) markers. Moreover, CXCR1, whose binder is CXCR1, was found as the main marker of ALDH^+^CSC [128]. This binding on the CSC surface somehow protected it from pro-apoptotic signals, and this was the rationale for the development of an allosteric inhibitor of CXCR1, reparixin, which was first tested in vitro in single administration or in combination with taxanes [129]. In 2017, reparixin was studied again in a pilot study combined with taxanes, showing some good results, especially in long-term observations [130], while in a more recent phase II trial (the fRida trial), a group of researchers from several different countries randomly assigned patients (in a 1:1 ratio) to a combined reparixin –paclitaxel treatment or placebo, without showing significant improvement in terms of PFS prolongation [131].

## 4. Focus on the New NCCN Clinical Practice Guidelines

In March 2023, the NCCN, the main association of cancer centers in the United States, released the updated clinical practice guidelines for the treatment of breast cancer. Among the aims of this review, there is a willingness to synthesize them in order to give readers a more precise point of view on the choices made by the NCCN in the setting of mTNBC.

Once the presence of a metastatic TNBC is confirmed, the NCCN guidelines underline that quantitative assessment of PD-L1 with the determination of the combined positive score (CPS) is necessary to proceed further with cancer classification. The CPS is calculated on the basis of the number of PD-L1^+^ cells (including tumor cells, lymphocytes, and macrophages) in relation to the total tumor cells and is one of the factors influencing the therapeutic choice [132].

As previously stated, immunotherapy has radically changed the approach of oncologists to cancers, and this is particularly true in the case of TNBC. In cases of recurrent unresectable (local or regional) or stage IV (M1) TNBC, the NCCN guidelines recommend both PD-L1 CPS and BRCA1/2 mutational status determination in the phase preceding the first treatment.

Table 1 summarizes all the main phase II and III trials concerning the FDA-approved drugs that are recommended in the most recent NCCN clinical practice guidelines for the treatment of recurrent unresectable or metastatic TNBC. We report all the trials that were specifically designed for TNBC, while all those considering other subtypes of BC were excluded.

In Table 2, instead, we recapitulate the recommended steps of therapy as suggested by the same NCCN guidelines.

When the PD-L1 CPS is more than or equal to 10, regardless of the *BRCA* mutation status, mTNBC is supposed to be responsive to combination therapy with pembrolizumab and systemic chemotherapy. On the contrary, in the case of a PD-L1 CPS lower than 10, the treatment choice depends on the germline *BRCA1/2* mutational status. In the case of mutated *BRCA1/2*, PARP inhibitors and platinum-based chemotherapy are the preferred regimens (both have a Category 1 recommendation, indicating uniform consensus in the intervention). When *BRCA1/2* mutations are absent, systemic chemotherapy becomes the preferred regimen. More details concerning this particular subgroup of treatment can be found in Table 3.

As for the second line of treatment, the choice must be taken depending on the *BRCA1/2* mutations, even if in any subtype of mTNBC, systemic chemotherapy (the same agents reported in Table 3) and sacituzumab govitecan can be considered for treatment (sacituzumab govitecan may be used for mTNBC patients who have received at least two prior therapies). In cases of *BRCA1/2* mutations, PARPis are again the preferred agents, while in cases with an absence of mutations and, simultaneously, are HER2 IHC 1^+^ or 2^+^/ISH-negative, the choice falls on fam-trastuzumab deruxtecan-nxki (anti-HER2).

According to the NCCN guidelines, in the case of third-line treatment, there are two main possibilities of treatment: systemic chemotherapy (again, the same agents reported in Table 3) or targeted agents based on specific genetic mutations. All third-line therapeutic agents are summarized in Table 4. Note that all of them are considered under Category 2A, meaning that their choice is based on a lower level of evidence.

Table 5 summarizes all the main mechanisms of action of the antineoplastic agents described above, considering only the categories of drugs approved by the FDA for mTNBC treatment.

## 5. Conclusions and Future Perspectives

Although immunotherapy has largely revolutionized the treatment of many types of tumors, including TNBC and its metastatic forms, chemotherapy still represents the treatment of choice for most mTNBC patients. New strategies considering different pathways are under consideration for improving the outcomes of the disease since there is still an unmet need for effective and precise treatments.

In this sense, precision therapy represents the approach chosen by the recent trials concerning anti-mTNBC drugs because of the significant burden of side effects caused by chemotherapeutic agents, but also because of their potential benefits in clinical practice. The extreme molecular and clinical variability of mTNBC has inspired the expansion of knowledge in the fields of genome-sequencing technology (where next-generation sequencing, NGS, is considered the method of choice), biomarker, and immunologic target investigation.

Various types of vaccines have been introduced in the last several years in the field of cancer prevention and treatment, and some of them have already been tested in preclinical studies concerning TNBC [133,134]. A recent review by Hosseini et al. [135] took stock of the evolving trends in this setting, concluding that, despite encouraging results in early trials, this strategy still needs to be explored in phase III randomized studies.

Similarly, the advances in viral genetic engineering have allowed for the development of oncolytic viruses able to recognize some cell receptors overexpressed in tumor tissues, but also to encode pro-apoptotic genes to deliver to cancer cells [136]. One study has recently focused on the role of one oncolytic virus in enhancing the functions of NK CD8+ T cells in the setting of TNBC, showing promising results [137], even though immune viral therapy still remains in its embryological stages and needs further exploration through randomized trials.

The advances in the field of metastatic TNBC have exponentially focused attention on patients’ quality of life since these subjects are often treated with several lines of antineoplastic agents and burdened with several side effects. These include early and late side effects, such as fatigue, alopecia, cytopenia, neurocognitive dysfunction or peripheral neuropathy, cardiomyopathy, and others, only considering the chemotherapeutic agents [138]. Diarrhea, infection, rash, hyperglycemia, and fatigue are commonly found in patients treated with kinase inhibitors [22,92,93], while dermatological reactions and autoimmune inflammatory diseases are frequently reported in patients treated with immune checkpoint inhibitors [139]. A lower burden of undesirable effects is usually registered, instead, in combined therapy regimens, which need to be preferred for their multiple different sites of action and the possibility of minimizing the adverse effects that higher doses of a single drug would cause.

Moreover, the oncologist cannot ignore that other medical conditions are frequently reported in mTNBC patients’ clinical history, and each of them could theoretically interfere in all phases of treatment. Diabetes and obesity, for example, were found to be particularly associated with a higher incidence of TNBC in the United States [140]. The findings concerning active smoking are controversial, while more certain results were reported in patients with chronic kidney disease (CKD) and renal replacement therapies, which seem to be strongly associated with TNBC development and deserve to be mentioned among the factors to consider for each step of therapy. Although neither CKD nor renal replacement therapy contraindicates surgery or radiotherapy, reduced renal function could affect the pharmacokinetics of drugs used in the systematic treatment to a different extent, increasing their toxicity and the risk of adverse drug reactions [141].

Although the research in the field of TNBC has moved forward in the last few years, it is clear how a great deal of progress is still needed. The high mortality rates caused by this subtype of breast cancer and the poor outcomes derived from metastases and/or relapses still require appropriate and specific therapies. To date, it seems that no single pharmacological therapy can contrast the multitude of clinical and molecular alterations induced by the cancer, while combination therapy with diverse agents could represent the opportunity for improving patients’ outcomes, with manageable safety profiles.

We are confident that research advances and both ongoing and future trials will further clarify other complicated aspects of TNBC and will provide new opportunities for contrasting one of the most aggressive forms of cancer.

Obviously, this study presents some limitations, and they are mainly related to the potential article selection biases. All articles were in fact screened by the authors, and we tried to minimize such biases with a double-blind screening; in addition, a third author was consulted in cases of eventual disagreement between the first two reviewers. Moreover, the research was limited to the two databases mentioned above (PubMed and Scopus) and not extended to other databases. Finally, the data cover a period of five years; this was the authors’ choice and it was based on the fact that the last five years indicatively represent the keystone in the field of mTNBC. Additionally, in the last several years, many systematic reviews (with or without meta-analyses) have been published on the same topics, and in our opinion, more extensive research would not have led to better conclusions. We still believe that, although, on one hand, this could represent a limitation of our findings, at the same time, it has allowed us to screen a huge number of studies and findings in the field of mTNBC.

## Figures and Tables

**Figure 1 biomedicines-11-01772-f001:**
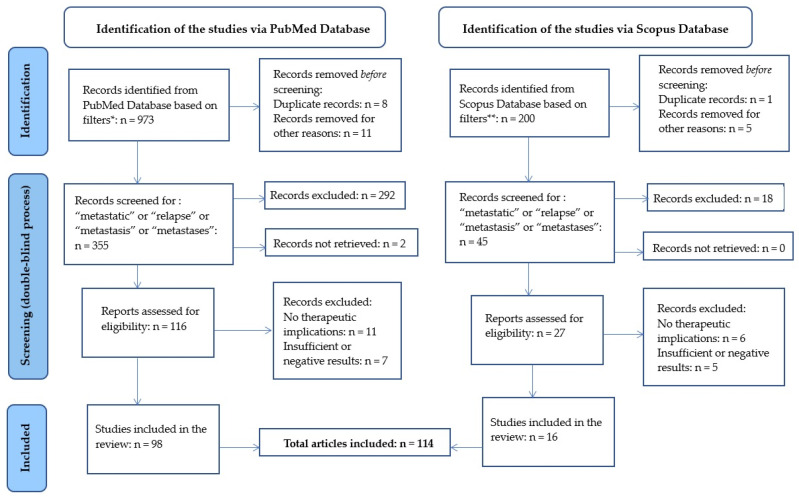
Comprehensive review flow diagram. * Filters applied for PubMed: “triple-negative breast cancer” or “TNBC” or “basal-like”, multicenter study, observational study, clinical study, clinical trial, (phase I, II, III, or IV), comparative study, review, systematic review, guideline, meta-analysis, preprints, studies involving adult patients. Publication years: 2018–2022, executed on 1 March 2023. ** Filters applied for Scopus: TITLE-ABS-KEY (“triple-negative” OR “triple-negative”) AND (“TNBC”) AND (“basal-like”), human. Publication years: 2018–2022, executed on 1 March 2023.

**Figure 2 biomedicines-11-01772-f002:**
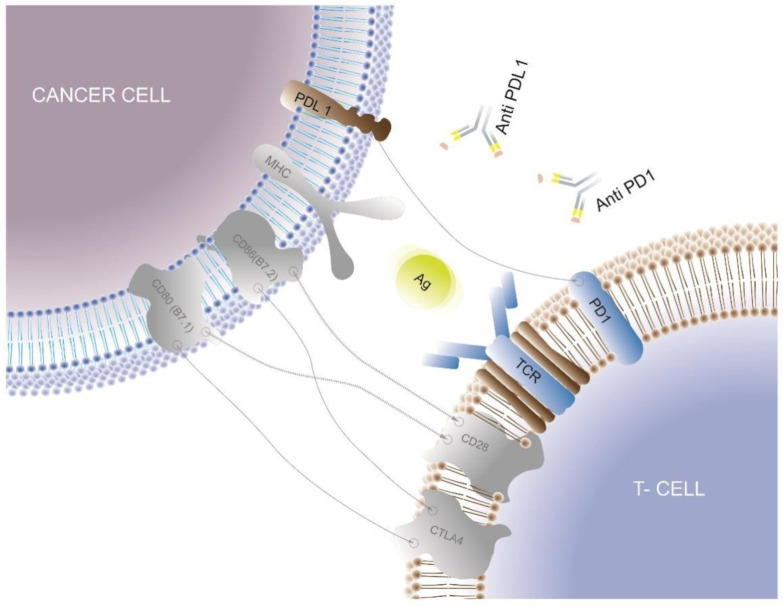
Interactions between the cancer cell and the T-cell: PD-1, located on the surface of the T-cell, binds PD-L1, on the surface of the cancer cell. Anti-PD-L1 and anti-PD-1 are able to inhibit this interaction.

**Figure 3 biomedicines-11-01772-f003:**
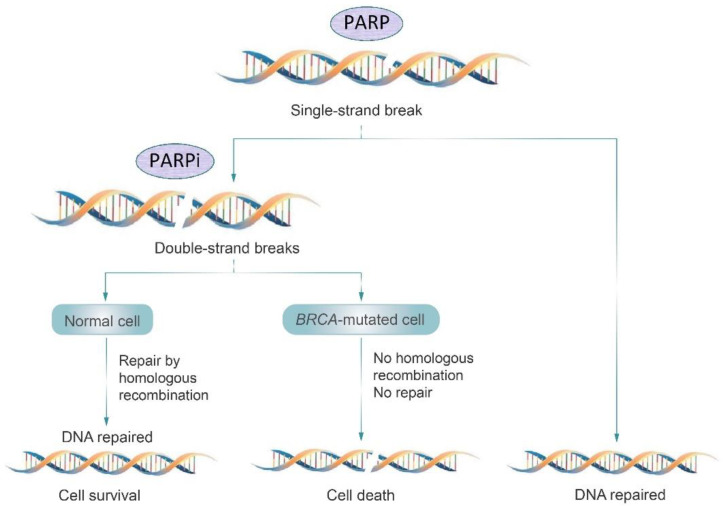
Mechanism of action of PARP inhibitors (PARPi). PARPi inhibits the PARP-mediated error-free repair of single-strand breaks (SSB), resulting in synthetic lethality in BRCA-associated cancer cells.

**Table 1 biomedicines-11-01772-t001:** Main phase II and III trials concerning specific metastatic TNBC treatments.

Authors	Trial Registration Code	Sample Size, *n*	Study Design	Study Type	Study Duration	Treatment	PFS and/or OS	Other Clinical Results
Zhang et al. (CBCSG006 trial) [14]	NCT01287624	240	Phase III	RCT	11 January–13 November	Cisplatin + gemcitabine (GP) vs. paclitaxel + gemcitabine (GT)	PFS: 7.73 (95% CI 6.46–9.00) for GP vs. 6.07 (95% CI 5.32–6.83) for GT arm OS: NS	-
Yardley et al. (tnAcity trial) [23]	NCT01881230	191	Phase II	RCT	13 September–15 April	Nab-paclitaxel + carboplatin (nab-P/C) vs. nab-paclitaxel + gemcitabine (nab-P/G) vs. gemcitabine + carboplatin (G/C)	PFS: nab-P/C vs. nab-P/G 8.3 vs. 5.5 (HR 0.59, 95% CI 0.38–0.92); nab-P/C vs. G/C 8.3 vs. 6.0 (HR 0.58, 95% CI 0.37–0.90) OS: NS	ORR: 73% for nab-P/C, 39% for nag-P/G, 44% for G/C
Wang et al. (GAP trial) [24]	NCT02546934	254	Phase III	RCT	16 March–19 October	Nab-paclitaxel + cisplatin (AP) vs. gemcitabine + cisplatin (GP)	PFS: 9.8 for AP vs. 7.4 for GP OS: HR 0.62 (95% CI 0.44–0.90) in favor of AP	ORR: 81.1% for AP vs. 56.3% for GP
Cortes et al. (EMBRACE trial) [36]	NCT00388726	762	Phase III	RCT	6–8 November	Eribulin mesilate (EM) vs. treatment of physician’s choice (TPC)	OS: 13.1 (95% CI 11.8–14.3) vs. 10.6 (95% CI 9.3–12.5); HR 0.81 (95% CI 0.66–0.99) in favor of EM arm	-
Anders et al. (LCCC 1525 trial) [47]	NCT02768701	40	Phase II, single arm	SAT	16 November–18 February	Cyclophosphamide prior to pembrolizumab	PFS: 1.8 months	ORR: 21%
Bardia et al. (ASCENT trial) [50]	NCT02574455	468	Phase III	RCT	17 November–19 September	Sacituzumab govitecan (SG) vs. chemotherapy (CT)	PFS: 5.6 (95% CI 4.3–6.3) for SG arm vs. 1.7 (95% CI 1.5–2.6) for CT arm OS: 12.1 (95% 10.7–14.0) for SG vs. 6.7 (95% CI 5.8–7.7) for CT arm	ORR: 35% for SG vs. 5% for CT arm
Schmid et al. (IMpassion130 trial) [60]	NCT02425891	902	Phase III	RCT	15 June–17 May	Atezolizumab + nab-paclitaxel (AT-P) vs. placebo + nab-paclitaxel (PP)	PFS: 7.5 vs. 5.0; HR 0.62 (95% CI 0.49–0.78) in favor of AT-P arm OS: NS	-
Røssevold et al. (ALICE trial) [62]	NCT03164993	70	Phase IIb	RCT	17 August–21 December	Atezolizumab + pegylated liposomal doxorubin + cyclophosphamide (AT-CT) vs. placebo + doxorubicin + cyclophosphamide (P-CT)	PFS: 4.3 vs. 3.5; HR 0.57 (95% CI 0.33–0.99) in favor of AT-CT arm	-
Miles et al. (IMpassion131 trial) [63]	NCT03125902	651	Phase III	RCT	17 August–19 September	Atezolizumab + paclitaxel (AT-P) vs. placebo + paclitaxel (PP)	PFS: 6.0 for AT-P arm vs. 5.7 for PP arm; HR 0.82 (95% CI 0.60–1.12) OS: NS	-
Adams et al. (KEYNOTE-086 trial) [64]	NCT02447003	170	Phase II, single arm	SAT	15 July–16 January	Pembrolizumab	PFS: 2.0 (95% CI 1.9–2.0) OS: 9.0 (95% CI 7.6–11.2)	ORR: 5.3% in the total and 5.7% in the PD-L1^+^ population
Ho et al. [65]	NCT02730130	17	Phase II, single arm	SAT	16 June–17 May	Pembrolizumab + radiotherapy	PFS: 2.6 months OS: 8.25 months	ORR: 17.6% (95% CI 4.7–44.2)
Winer et al. (KEYNOTE-119 trial) [67]	NCT02555657	622	Phase III	RCT	15 November–17 April	Pembrolizumab (Pem) vs. chemotherapy of physician’s choice (TPC)	OS: 9.9 (95% CI 8.3–11.4) for Pem arm and 10.2 (95% CI 7.9–12.6) for TPC arm	-
Cortes et al. (KEYNOTE-355 trial) [68]	NCT02819518	847	Phase III	RCT	17 January–18 June	Pembrolizumab (Pem) vs. chemotherapy treatment of physician’s choice (TPC)	OS: 23.0 for Pem arm (with PD-L1 CPS ≥10) vs. 16.1 for TPC arm; HR 0.86 (95% CI 0.72–1.04)	-
Tan et al. [87]	NCT02978716	34	Phase II	RCT	17 February–18 May	Trilaciclib (Tr) prior to gemcitabine + carboplatin (G/C) vs. G/C alone	OS: 12.6 for G/C arm, not reached for Tr prior to G/C arm, and 19.8 for Tr + Tr prior to G/C arm	-
Schmid et al. (PAKT trial) [22]	NCT02423603	140	Phase II	RCT	14 May–17 June	Capivasertib + paclitaxel (CP) vs. placebo + paclitaxel (PP)	PFS: 5.9 for CP arm vs. 4.2 for PP arm, HR 0.74 (95% CI 0.50–1.08) OS: 19.1 for CP arm vs. 12.6 for PP arm, HR 0.61 (95% CI 0.37–0.99)	-
Garrido-Castro et al. [92]	NCT01790932	50	Phase II, single arm	SAT	12 June–14 September	Buparlisib	PFS: 1.8 (95% CI 1.6–2.3) OS: 11.2 (95% CI 6.2–25)	-
Lehmann et al. (TBCRC 032 trial) [93]	NCT02457910	17	Phase IB/II	RCT	15 May–18 August	Enzalutamide alone (Ez) vs. enzalutamide + taselisib (Ez/T)	PFS: 3.4 months	CBR: 35.7%
Diamond et al. [95]	NCT01639248	41	Phase II, single arm	SAT	12 July–16 October	ENMD-2076	PFS: 1.84 (95% CI 1.73–3.73)	CBR: 16.7% (95% CI 6–32.8%)
Brufsky et al. (COLET trial) [98]	NCT02322814	106	Phase II	RCT	15 March–16 October	Cobimetinib + paclitaxel (CoP) vs. placebo + paclitaxel (PP)	PFS: 5.5 for CoP arm vs. 3.8 for PP arm, HR 0.73 (95% CI 0.43–1.24)	ORR: 38.3% (95% CI 24.4–52.2) for CoP arm vs. 20.9% (95% CI 8.8–33.1) for PP arm
Goldstein et al. (fRIDA trial) [131]	NCT01861054	123	Phase II	RCT	15 July–18 May	Reparixin + paclitaxel (RP) vs. placebo + paclitaxel (PP)	PFS: 5.5 for RP arm vs. 5.6 vs. PP arm	-

PFS = progression-free survival, ORR = objective response rate; OS = overall survival; NS = not significant; CBR = clinical benefit rate; RCT = randomized controlled trial; SAT = single-arm trial.

**Table 2 biomedicines-11-01772-t002:** Systemic therapy regimens for recurrent unresectable or metastatic (stage IV) disease (NCCN Guidelines Version 4.2023).

Setting	Subtype/Biomarkers	Regimen
First-line	PD-L1 CPS ≥ 10 regardless of germline BRCA mutation	Pembrolizumab + chemotherapy (nab-paclitaxel, or gemcitabine and carboplatin) (Category 1)
	PD-L1 CPS < 10 and no germline BRCA1/2 mutation	Systemic chemotherapy (see Table 3)
	PD-L1 CPS < 10 and germline BRCA1/2 mutation	PARPi (olaparib, talazoparib) or platinum (carboplatin or cisplatin) (both Category 1)
Second-line	Germline BRCA1/2 mutation	PARPi (olaparib, talazoparib) (Category 1)
	Any	Sacituzumab govitecan (Category 1)
	No germline BRCA1/2 mutation and HER2 IHC 1+ or 2 + /ISH negative	Fam-trastuzumab deruxtecan-nxki (Category 1)
Third-line and beyond	Biomarker positive	Targeted agents (see Table 4)
	Any	Systemic chemotherapy (see Table 3)

**Table 3 biomedicines-11-01772-t003:** Systemic chemotherapy for HR-positive or negative and HER2-negative breast cancer (NCCN Guidelines Version 4.2023).

Preferred Regimens	Other Recommended Regimens	Useful in Certain Circumstances
Anthracyclines (doxorubicin or liposomal doxorubicin)	Cyclophosphamide	AC (doxorubicin and cyclophosphamide)
Taxanes (paclitaxel)	Docetaxel	EC (epirubicin and cyclophosphamide)
Anti-metabolites (capecitabine or gemcitabine)	Nab-paclitaxel	CMF (cyclophosphamide and methotrexate and fluorouracil)
Microtubule inhibitors (vinorelbine or eribulin)	Epirubicin	Docetaxel and capecitabine
	Ixabepilone	GT (gemcitabine and paclitaxel)
		Carboplatin and paclitaxel or nab-paclitaxel
		Gemcitabine and carboplatin

**Table 4 biomedicines-11-01772-t004:** Additional targeted therapies and associated biomarker testing for recurrent unresectable or metastatic (stage IV) disease (NCCN Guidelines Version 4.2023).

Biomarker	Detection Method	FDA-Approved Agents
NTRK fusion	FISH, NGS, PCR (tissue block)	Larotrectinib or entrectinib (Category 2A)
MSI-H/dMMR	IHC, NGS, PCR (tissue block)	Pembrolizumab or dostarlimab-gxly (Category 2A)
TMB-H (≥ 10 mut/Mb)	NGS	Pembrolizumab (Category 2A)
RET-fusion	NGS	Selpercatinib (Category 2A)

FISH = fluorescent in situ hybridization; NGS = next-generation sequencing; PCR = polymerase chain reaction; TMB-H = tumor mutational burden—high; mut/Mb = mutation per megabase.

**Table 5 biomedicines-11-01772-t005:** Mechanisms of action of the main antineoplastic agents approved for metastatic TNBC.

Antineoplastic Agents	Main Mechanism of Action
Chemotherapeutic agents	
Platinum-based chemotherapy	Covalent binding to DNA, leading to the formation of DNA cross-links
Taxanes	Binding to microtubules, preventing their depolymerization
Anthracyclines	Disruption of DNA by poisoning topoisomerase
Microtubule inhibitors	Inhibition of the AKT/mTOR signaling pathway
Alkylating agents	Direct action on DNA, resulting in crosslinking and strand breaks
Antibody–drug conjugates	Delivery of deactivated cytotoxins to specific cancer cells
Immune checkpoint inhibitors	Targeted inhibition of the bindings between cancer cell checkpoint ligands and their complementary receptors on the CD8+ cell
PARP inhibitors	Inhibition of DNA repair pathways, leading to apoptosis of cancer cells, especially in homologous recombination deficient cells
Tyrosine kinase inhibitors	Phosphorylation of specific amino acids on substrate enzymes, causing altered signal transduction

## Data Availability

The datasets generated and/or analyzed during the current study are not publicly available but are available from the corresponding author upon reasonable request.

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
