# Peer review of "Update on Classic and Novel Approaches in Metastatic Triple-Negative Breast Cancer Treatment: A Comprehensive Review"

_biomedicines, 2023, doi:10.3390/biomedicines11061772_

Round 1

Reviewer 1 Report

1.     In Figure 1, it shows that 995 articles have been initially retrieved from PubMed and 200 from Scopus, but in Line 131, authors mentioned 973 articles from PubMed and 231 articles from Scopus. Why this discrepancy in numbers?

2.     Why negative results have been excluded from the analysis as indicated in Figure 1? This will create biasedness in the results.

3.     What is the reason for limiting the search to last 5 years?

4.     In Line 160, it should be SG and NF, not SG and NG.

5.     At all places, authors are claiming that they used PubMed and Scopus databases, but in line 164, they added Medline too.

6.     The way the manuscript text is underlined and coloured, it shows that it is not a fresh manuscript, but a revised manuscript as per reviewer’s responses.

7.     In the manuscript, at some places, authors have italicized the drugs name to highlight and bring a focus to them, but on other places, drug names are non-italicized. Authors should maintain uniformity.

8.     Section 3 should contain novel tables and figures to make the data more representable.

9.     In Table 1, authors should add two columns, one is the Study type, and second is about the duration of the study.

10.   What are the limitations of present study? It should be mentioned in Conclusion section. 

Minor corrections only

Author Response

We want to thank the reviewer for his/her comments. We think that each of them are reasonable and we hope he/she could appreciate our efforts.

  1. In Figure 1, it shows that 995 articles have been initially retrieved from PubMed and 200 from Scopus, but in Line 131, authors mentioned 973 articles from PubMed and 231 articles from Scopus. Why this discrepancy in numbers?

Thanks for pointing this out. The correct number of articles retrieved from Pubmed was 973 as stated in line 131; 200 was the correct number for Scopus articles, instead. We modified it in the article and in Figure 1.

  1. Why negative results have been excluded from the analysis as indicated in Figure 1? This will create biasedness in the results.

The aim of our systematic review was to screen the articles showing targeted therapies in the field of mTNBC with promising or favourable results. That is why we made the articles’ selection excluding those studies with negative results.

  1. What is the reason for limiting the search to last 5 years?

This was our choice and it was based on the fact that cancer treatment research is in continuous evolution and the last four-five years have represented a real keystone in the field of mTNBC. Moreover, in the previous years, many systematic reviews and meta-analyses were published on the same topics and a more time-extended research would have not taken us to better conclusions. With this 5-year article research, we were able to screen a huge number of articles concerning mTNBC anyway.

  1. In Line 160, it should be SG and NF, not SG and NG.

Thanks for noticing this, it was immediately corrected.

  1. At all places, authors are claiming that they used PubMed and Scopus databases, but in line 164, they added Medline too.

That is true, it was our mistake. Medline database was not used for article screening; we removed it from the manuscript.

  1. The way the manuscript text is underlined and coloured, it shows that it is not a fresh manuscript, but a revised manuscript as per reviewer’s responses.

Sorry for the inconvenient; this was a clear mistake in the phase of preparation of the manuscript. We promptly removed underlined words and colours.

  1. In the manuscript, at some places, authors have italicized the drugs name to highlight and bring a focus to them, but on other places, drug names are non-italicized. Authors should maintain uniformity.

We decided to italicize only those drugs that were included in the last NCCN guidelines for mTNBC treatment

  1. Section 3 should contain novel tables and figures to make the data more representable.

Novel figures were added to this section and a summarizing table with all mechanisms of action was added in the text (table 5).

  1. In Table 1, authors should add two columns, one is the Study type, and second is about the duration of the study.

Both columns were added to table 1. Thanks for this suggestion.

  1. What are the limitations of present study? It should be mentioned in Conclusion section. 

We mentioned all the limitations in the last section, as correctly suggested by the reviewer.

Reviewer 2 Report

In the present paper, the authors have summarized the current therapeutic strategies to treat triple negative breast cancer (TNBC). This review article is very comprehensive and adequate. The manuscript is well written and easy to follow. The authors provided summarizing tables that helps the readability of the paper. Conclusions and future perspectives are highlighted. I have no major comments/concerns. The paper can be accepted in its current form.

Author Response

Many hanks for your kind review; it was really appreciated by all authors.

Reviewer 3 Report

Specific comments to the authors

The authors Salvatore Greco et al. of the submitted review "Update on classic and novel approaches in metastatic Negative Breast Cancer treatment: a comprehensive review" collect, summarise and analyse heterogeneous therapeutic aspects of triple negative breast cancer (TNBC) based on already published in vitro and in vivo experiments as well as clinical trials. To this end, the authors performed an intensive search of the pubmed and Scopus databases.

The topics presented range from epidemiology, histomorphological/molecular findings/characteristics of breast cancer to therapeutic options for TNBC in the context of chemotherapy, immune checkpoint inhibitors, PARP inhibitors, thyrosine kinase inhibitors and alternative modalities. In summary, the author provides an up-to-date overview of possible treatment options for this aggressive breast tumour entity that is mostly easy to read, follow and understand. The authors should clarify some aspects before accepting the manuscript for publication, as mentioned below.

# Abstract: Authors should state the procedure used to obtain the data for the comprehensive review.

Introduction: Please provide the link "www.who.int" and "www.cancer.org" in relation to the topic of TNBC. The basic clinical, pathological and molecular characteristics of the breast cancer subtypes could be included in a separate table as an overview.

# Table 1: The results of the clinical endpoints should be presented in separate columns. The authors should interpret these clinical endpoints together.

# Conclusions and future perspectives: This section should include a paragraph on "how I would treat a patient with TNBC" based on the published clinical trials.

Minor editing of English language required.

Author Response

The authors Salvatore Greco et al. of the submitted review "Update on classic and novel approaches in metastatic Negative Breast Cancer treatment: a comprehensive review" collect, summarise and analyse heterogeneous therapeutic aspects of triple negative breast cancer (TNBC) based on already published in vitro and in vivo experiments as well as clinical trials. To this end, the authors performed an intensive search of the pubmed and Scopus databases.

The topics presented range from epidemiology, histomorphological/molecular findings/characteristics of breast cancer to therapeutic options for TNBC in the context of chemotherapy, immune checkpoint inhibitors, PARP inhibitors, thyrosine kinase inhibitors and alternative modalities. In summary, the author provides an up-to-date overview of possible treatment options for this aggressive breast tumour entity that is mostly easy to read, follow and understand. The authors should clarify some aspects before accepting the manuscript for publication, as mentioned below.

Abstract: Authors should state the procedure used to obtain the data for the comprehensive review.

We searched the PubMed and Scopus databases for free-full-text reported in the last 5-year literature including the words “triple negative breast cancer” or “TNBC” or “basal-like”. The articles were analyzed by the authors independently and double-blindly and a total of 114 articles were included in the review.

Introduction: Please provide the link "www.who.int" and "www.cancer.org" in relation to the topic of TNBC.

Both links were correctly replaced.

The basic clinical, pathological and molecular characteristics of the breast cancer subtypes could be included in a separate table as an overview.

The supplementary figure 1 was created following some of the suggestions of the reviewer, in order to clarify the molecular characteristics of the various subtypes of TNBC according to PAM50 and Lehmann’s classifications. Since the aim of this review is to discuss about the new possible targets of therapy against TNBC we thought not to add further details concerning the different subtypes of TNBC.

Table 1: The results of the clinical endpoints should be presented in separate columns. The authors should interpret these clinical endpoints together.

We created two different columns: in the first one we put PFS and OS, while in the second one all other clinical results.

Conclusions and future perspectives: This section should include a paragraph on "how I would treat a patient with TNBC" based on the published clinical trials."

This section is already present in the chapter 4. “Focus on the new NCCN clinical practice guidelines”, where the NCCN guidelines are “slavishly” followed. A new paragraph in the last chapter would have meant a series of redundant informations.

The authors thank the reviewer for his/her kind comments about our article. We did our best to respond point-by-point to each concern, although some of these could not be totally fulfilled.

Reviewer 4 Report

This paper systematically reviewed the last findings in the setting of metastatic Triple Negative Breast Cancer (TNBC) treatment, focusing on each category of drugs approved by FDA and included in the NCCN guidelines. In addition, the last published studies was introduced, which reported new and promising molecules able to specifically target some of the biomarkers involved in TNBC pathogenesis. The topic fits the scope of this journal, and may benefit the development of treatment against TNBC pathogenesis. In general, the manuscript is well-organized, and the references can support the conclusions. The key issues should be addressed before its publication on Biomedicines.

1. The underlines of the key points are suggested to be removed.

2. The cell signaling pathways related to the treatment options are required to be introduced with a figure legend.

The English language is good.

Author Response

Thanks for your comments. Attached you can find the point-by-point responses to your review.

Round 2

Reviewer 1 Report

The authors have carefully revised the manuscript and addressed most of my concerns. However, I would like to suggest authors revise the manuscript as per the suggestion given below to make it more representable.

1.   In section 3, the Authors have added a new figure for PARP inhibitors. Authors should prepare other figures to cover other sub-sections of section 3.

Author Response

Dear reviewer,

Thank you for your efforts in improving our paper. You wrote: “The authors have carefully revised the manuscript and addressed most of my concerns. However, I would like to suggest authors revise the manuscript as per the suggestion given below to make it more representable.1.     In section 3, the Authors have added a new figure for PARP inhibitors. Authors should prepare other figures to cover other sub-sections of section 3."We disagree with you, adding figures will not improve the text, considering that a summary image has been added in the supplementary material. We do not want to annoy you, not to reduce the value of your advice, however to add so many figures will take a lot of effort and resources to our groups, considering that the text is clear enough.

Thank you for your time and clever suggestions.